# Degradation, Bone Regeneration and Tissue Response of an Innovative Volume Stable Magnesium-Supported GBR/GTR Barrier Membrane

**DOI:** 10.3390/ijms21093098

**Published:** 2020-04-28

**Authors:** Mike Barbeck, Lennart Kühnel, Frank Witte, Jens Pissarek, Clarissa Precht, Xin Xiong, Rumen Krastev, Nils Wegner, Frank Walther, Ole Jung

**Affiliations:** 1Department of Oral Maxillofacial Surgery, Division of Regenerative Orofacial Medicine, Study Group: Biomaterials/Surfaces, University Medical Center Hamburg-Eppendorf, 20246 Hamburg, Germany; 2BerlinAnalytix GmbH, 12109 Berlin, Germany; 3Biotrics Bioimplants GmbH, 12109 Berlin, Germany; 4NMI Natural and Medical Sciences Institute, University of Tübingen, 72770 Reutlingen, Germany; 5Faculty of Applied Chemistry, Reutlingen University, 72762 Reutlingen, Germany; 6Department of Materials Test Engineering (WPT), TU Dortmund University, 44227 Dortmund, Germany; 7Clinic and Policlinic for Dermatology and Venereology, University Medical Center Rostock, 18057 Rostock, Germany

**Keywords:** barrier membrane, GBR/GTR membrane, degradation, magnesium, collagen, in vivo, histomorphometry, tissue reaction

## Abstract

Introduction: Bioresorbable collagenous barrier membranes are used to prevent premature soft tissue ingrowth and to allow bone regeneration. For volume stable indications, only non-absorbable synthetic materials are available. This study investigates a new bioresorbable hydrofluoric acid (HF)-treated magnesium (Mg) mesh in a native collagen membrane for volume stable situations. Materials and Methods: HF-treated and untreated Mg were compared in direct and indirect cytocompatibility assays. In vivo, 18 New Zealand White Rabbits received each four 8 mm calvarial defects and were divided into four groups: (a) HF-treated Mg mesh/collagen membrane, (b) untreated Mg mesh/collagen membrane (c) collagen membrane and (d) sham operation. After 6, 12 and 18 weeks, Mg degradation and bone regeneration was measured using radiological and histological methods. Results: In vitro, HF-treated Mg showed higher cytocompatibility. Histopathologically, HF-Mg prevented gas cavities and was degraded by mononuclear cells via phagocytosis up to 12 weeks. Untreated Mg showed partially significant more gas cavities and a fibrous tissue reaction. Bone regeneration was not significantly different between all groups. Discussion and Conclusions: HF-Mg meshes embedded in native collagen membranes represent a volume stable and biocompatible alternative to the non-absorbable synthetic materials. HF-Mg shows less corrosion and is degraded by phagocytosis. However, the application of membranes did not result in higher bone regeneration.

## 1. Introduction

Bony defects, resulting from trauma, tumor resection, malformations or progressive degeneration, often require difficult reconstructive bone augmentation procedures. The application of guided bone-/guided tissue-regeneration (GBR/GTR) technique is a common method to realize bone regeneration in the alveolar ridge area [1,2,3]. The main principle of GBR/GTR is based on a barrier membrane between the defect and outer soft tissue to prevent rapid soft tissue ingrowth and to promote osseous regeneration [4,5,6,7,8,9]. In this context, a barrier membrane has to fulfil the five postulated criteria of Scantlebury: tissue integration, cell occlusivity, clinical manageability, space-making and biocompatibility [10].

For dental application, resorbable and non-resorbable membranes of different origins can be distinguished [8]. Furthermore, natural and synthetic materials are available [8]. For indications that do not require volume stability, collagen-based membranes have shown to be optimal biomaterials due to their biocompatibility and degradability that are conducted by cells of the collagen metabolism such as fibroblasts, macrophages and eosinophils [11,12,13,14]. However, conventional collagen membranes are usually unstable and can either collapse into the defect or deform under load, resulting in the impairment of tissue regeneration [12,13]. Thus, multidimensional bony defects require the application of volume-stable load-bearing membranes [15,16,17]. For this indication, non-resorbable materials based on polytetrafluorethylene (e-PTFE) or dense polytetrafluorethylene (d-PTFE) combined with titanium meshes are the gold standard [18,19]. However, the use of non-resorbable materials always requires a second surgical procedure for removement, which can comprise higher overall morbidities, patient discomfort and higher costs [12,13,19]. Besides slightly increased inflammation, the use of non-resorbable membranes can cause bacterial colonization in the event of oral exposure with concomitant regeneration impairment [20,21,22,23].

Thus, the clinical need to develop a next generation of volume-stable barrier membranes that combine structural stability with bioresorbable properties is of great interest. Based on the non-resorbable PTFE membranes with titanium meshes, the development of a resorbable collagen membrane with an integrated resorbable metallic mesh is obvious.

In this context, magnesium and a variety of its alloys have been a subject of recent investigations as a degradable implant material [24,25,26]. Beside its favorable bone-like properties, magnesium is fully degradable in aqueous solutions and biocompatible in natural doses [25,27,28,29]. However, the rapid uncontrolled degradation along with hydrogen gas evolution, environmental alkalization and rapid structural integrity loss can lead to necrosis and emphysema, and therefore limits a clinical application so far [25,30]. A variety of strategies have been developed to control magnesium degradation and to improve biocompatibility (Figure 1) [31]. Thereby, passivation of the surface can be achieved by conversion (e.g., plasma electrolytic oxidation, ion implantation) and deposited coatings (e.g., chemical vapor deposition, plasma spraying) of the surface [24,25,26,31,32].

As part of the conversion coatings, hydrofluoric acid (HF) treatment is a promising surface passivation method [33,34,35,36,37]. Thereby, hydroxides, oxides and compounds of the general formula Mg(OH)_x_F_2−x_ are formed on the magnesium surface during processing [37,38]. Interestingly, the corrosion rates were up to 20 times lower than those of untreated samples [37,38].

Considering these aspects, the present study aims to investigate a newly developed volume-stable barrier membrane composed of a native collagen membrane with an integrated magnesium mesh for GBR/GTR treatment. HF-treatment of magnesium was applied for passivation aspects and to slow down biodegradation. The newly developed membrane was investigated by established in vitro methods and an in vivo calvarian implantation model using New Zealand White Rabbits [39,40]. Tissue regeneration and reaction were analyzed using established histological, histopathological and histomorphometrical methods [19,41,42,43].

## 2. Results

### 2.1. Manufacturing of a Magnesium Collagen Membrane

Both in vitro and in vivo test samples could be successfully manufactured. Figure 2 displays the HF-treated and untreated magnesium sheets and meshes. The HF-coated samples are characterized by a matte and textured surface, while the untreated sheets have a more brilliant surface. Figure 3 shows the integration of the mesh into the collagen membrane as well as its formability and adaptability.

### 2.2. Cytocompatibility Analysis

Generally, values >70% of the negative control in the XTT and BrdU assays and values <130% of the negative control in the LDH assay indicate the nontoxic range as defined in ISO 10993-5:2009 [29]. In both the XTT and BrdU assays, HF-treated magnesium exhibited suitable cytocompatibility, whereas the untreated magnesium was within the toxic range (Figure 4A–C). Thereby, HF-treated magnesium was not significantly different compared to the negative control in the BrdU assay (*p* > 0.05), but significantly different from the negative control in the XTT assay (*p* ≤ 0.05). Both the untreated magnesium and positive control groups were significantly different to the negative control (*p* ≤ 0.01). In the LDH assay, both magnesium groups were inside the toxic range, while only the HF-treated magnesium group was not significantly different compared to the negative control (*p* > 0.05). Morphological assessment of the magnesium test samples after 72 h of extraction showed an enhanced corrosion of the untreated test samples, while the HF-treated specimens showed no visible signs of corrosion or degradation (Figure 4D)

In the live-dead staining assays, the positive and negative controls showed the expected qualitative direct cell responses. Thereby, green fluorescence indicates living cells, while red fluorescence indicates dead cells. Good cell attachment onto the surface is displayed by spindle-shaped cell morphology, while a round morphology suggests a lack of adhesion. Overall, the HF-treated group showed similar results to those of the negative control, although the attachment of the MC3T3 even appeared to be improved (Figure 5). Untreated magnesium exhibited poor cell attachment with some red dead cells. None of the magnesium specimens showed gas bubble dissolution in the direct test.

### 2.3. Histopathological Results

All 18 test animals completed the study. No signs of infection or wound dehiscence could be observed.

At 6 weeks post implantation, the histological analyses showed that comparable amounts of newly formed bone were found within the former defect areas of all study groups (Figure 8D). At this study time point, only remnants of the magnesium meshes in the respective study groups were detectable (Figure 6 and Figure 7). Moreover, no histological signs of the collagen membranes have been detected in these groups. Additionally, in the study group of the pure collagen membrane, no histological signs of the biomaterial were visible. In the sham operation group, only within the central regions of the former defect areas a thin layer of dense vessel- and cell-poor connective tissue, including mostly fibroblasts and some single macrophages, were detected at this earliest study time point.

Clear differences in the tissue responses to the HF-treated and the untreated magnesium materials were histologically observable (Figure 7). In the case of the untreated magnesium meshes, the mesh bars were surrounded by distinctly pronounced gas cavities. The magnesium bars were only connected to the surrounding connective tissue in small contact areas. Both the material- and the cavity-adherent connective tissue showed signs of a slight fibrosis involving mostly fibroblasts. Within the neighbored connective tissue, only moderate correlates of a material-associated inflammatory response, including mostly fibroblasts and low numbers of macrophages, have been detected at this study time point without signs of a phagocytic activity.

In the group of the HF-treated magnesium meshes, histological signs of less pronounced gas cavities outgoing from the magnesium meshes could be detected (Figure 8A). Moreover, the tissue reaction to this material included a thin layer of phagocyting cells at the implant-tissue surfaces (Figure 7A,C,E). This cell layer was furthermore surrounded by a dense fibrotic capsule as also observed in the group of the untreated meshes (Figure 7B,D,E). Despite the observed different tissue responses and the magnesium-associated gas cavities, the magnesium meshes in both study groups were most of the in close contact to newly formed bone tissue (Figure 7).

At 8 weeks post implantation, still comparable amounts of newly formed bone tissue were observable in all four study groups and the amounts did visibly not differ from that observed at 6 weeks post implantation. At this study time point, still remnants of the magnesium meshes in the both respective study groups were detectable (Figure 6 and Figure 7). The size of the material-related gas cavities in the group of the untreated magnesium meshes has visibly been decreased, while their peculiarity in the group of the HF-treated materials was comparably to the sizes at 6 weeks post implantation (Figure 8A). Moreover, the tissue responses were still comparable to that at 6 weeks post implantation (Figure 7). Thus, a fibrosis has been observed in case of the untreated magnesium meshes without any signs of phagocytic activities, while a thin cell layer of macrophages combined with a fibrosis-like capsule was found in the group of the HF-treated materials (Figure 7). No histological correlates of inflammatory processes were found in the surrounding connective tissue. In the group of the pure collagen membrane and the sham operation group, a thin layer of dense vessel- and cell-poor connective tissue including mostly fibroblasts were found only within the central regions of the former defect areas at this time point.

At 12 weeks post implantation, still comparable amounts of newly formed bone were found in all study groups and no differences that were additionally visibly comparable to the amounts at the former two study time points (Figure 8D). At this study time point, only very small remnants of the magnesium meshes could be microscopically detected (Figure 6 and Figure 7). Interestingly, the tissue responses to the grids were equivalent at this latest study time point (Figure 7). In both groups, i.e., the group of the untreated magnesium meshes and the HF-treated material, no histological signs of gas cavities have been observed any longer and the remnants of both materials were surrounded by a thin cell layer, which was composed of single fibroblasts (Figure 7). No signs of a material-associated fibrosis or attached phagocytes could be observed at this time point. No signs of inflammatory processes were detected in the surrounding connective tissue. Furthermore, the aforementioned layers of dense connective tissue within the central defect regions were still detectable in the group of the pure collagen membrane and the sham operation group.

### 2.4. Digital Volume Tomography (DVT) Analysis

Digital volume tomography (DVT) showed no statistically significant differences between the amounts of newly built bone in the four different study groups at all time points (Figure 8B–D and Figure 9). After 6 weeks, 76.17% ± 9.50% newly built bone was found in the group of the HF-treated magnesium mesh, which was comparable to the amounts of regenerated bone in the groups of the uncoated mesh (75.69% ± 9.23%), collagen group (77.86% ± 7.59%) and control group (74.25% ± 7.13%). At 12 weeks post implantation, the DVT analysis showed similar results, namely 72.17% ± 11.05% in the group of the HF-treated magnesium mesh and comparable 70.23% ± 7.49% in the groups of the uncoated mesh, the collagen group (77.11% ± 8.50%) and in the control group (70.98% ± 11.50%). At 18 weeks post implantation, equal results for the HF-treated magnesium mesh (75.50% ± 19.24%), the uncoated mesh (77.93% ± 9.94%), the collagen group (78.59% ± 13.88%) and the control group (78.58% ± 12.19%) were observed.

### 2.5. Contact Radiography

Overall, ossification increased over time in all study groups (Figure 8B–D and Figure 9). Thereby, the extent of ossification was lower compared to DVT. After 6 weeks, bone regeneration of more than 60% was noticeable in all groups, while the empty control group tend to ossify at a slower extent. However, this effect is not significant. At all time points, no significant differences between all study groups (*p* > 0.05) could be detected.

After 6 weeks, an amount of 62.37% ± 13.77% new built bone was found in the group of the HF-treated magnesium mesh, which was comparable to the amounts of rebuilt bone in the groups of the uncoated mesh (66.78% ± 5.11%), the collagen group (67.18% ± 14.29%) and in the control group without biomaterial insertion (55.64% ± 12.98%).

At 12 weeks post implantation, the DVT analysis showed similar results, namely 64.26% ± 16.42% in the group of the HF-treated magnesium mesh and comparable 69.45% ± 10.80% in the groups of the uncoated mesh, the collagen group (68.59% ± 12.90%) and in the control group (59.50% ± 19.61%).

At 18 weeks post implantation, equal results for the HF-treated magnesium mesh (74.07% ± 15.28%), the uncoated Mg mesh (69.85% ± 5.61%), the collagen group (69.15% ± 10.84%) and the control group (70.49% ± 11.96%) were observed.

### 2.6. Histomorphometrical Analysis

Connective tissue fillings as well as island-shaped ossifications in the defect area showed an increase over time for all groups. There was no indication of membrane collapse for neither of the membranes. The mean values of bone surface regeneration were clearly below those of DVT and contact radiography.

The quantitative measurements of bone regeneration using histomorphometrical analysis showed that no statistically significant differences between the amounts of newly regenerated bone were detectable within the four different study groups at 6 weeks post implantation (Figure 8D). Thus, an amount of 46.48% ± 11.42% new built bone was found in the group of the HF-treated magnesium mesh, which was comparable to the amounts of rebuild bone in the groups of the uncoated mesh (50.83% ± 22.09%), the collagen group (49.40% ± 17.55%) and the control group without biomaterial insertion (47.88% ± 22.34%).

At 12 weeks post implantation, the DVT analysis showed similar results, namely 41.21% ± 29.17% in the group of the HF-treated magnesium mesh and comparable 52.02% ± 19.55% in the groups of the uncoated mesh, the collagen group (46.59% ± 18.34%) and the control group (62.20% ± 20.86%).

At 18 weeks post implantation, equal results for the HF-treated magnesium mesh (56.59% ± 33.60%), the uncoated mesh (59.79% ± 18.14%), the collagen group (74.39% ± 14.40%) and the control group (58.15% ± 10.46%) were observed.

### 2.7. Results of the Gas Cavity Measurements

The quantitative measurements of gas cavity dimension via digitized total scans showed statistically significant differences between the HF-treated and uncoated magnesium meshes at 6 weeks post implantation. The surface of 535,453 ± 323,998 µm^2^ for uncoated magnesium showed to be significantly higher than the gas cavity dimension of 173,713 ± 129,697 µm^2^ found in the group of HF-coated meshes (*p* < 0.01) (Figure 8A).

At 12 weeks post implantation, the gas cavity measurements revealed lower results for the HF-treated (445,522 ± 44,457 µm^2^) compared to the uncoated magnesium meshes (131,299 ± 129,697 µm^2^). The reduction in cavity size for untreated magnesium meshes was found to be significant different compared to 6 weeks post implantation (*p* < 0.05).

Both magnesium meshes were associated with gas cavities, while they gradually decreased over time. Interestingly, the observed gas cavities did not interfere with bone regeneration. Moreover, the histomorphometrical analysis revealed that the size of gas cavities was significantly lower in the group of the HF-treated magnesium meshes up to 6 weeks (*p* < 0.01) after implantation. Furthermore, the gas cavity dimension of the untreated magnesium meshes reduced significantly after 12 weeks (*p* < 0.05). After 18 weeks, no gas cavities could be detected.

## 3. Discussion

In dentistry and other medical disciplines, bioresorbable GBR/GTR-membranes provide a valid treatment option for non-weight-bearing (bone) regeneration procedures, while special indications, such as high-volume defects, may require volume-stable membranes [12,13]. So far, the available volume-stable membranes consist of non-resorbable materials and require surgical removal after therapy. In this context, magnesium has manifoldly been shown to constitute a favorable biomaterial for the development of stabilizing structures due to its satisfying biocompatibility, biodegradability and mechanical properties [29,44,45,46]. As a promising surface modification for the delay of premature degradation, and thus, hydrogen release, hydrofluoric acid (HF)-treatments and the resulting magnesium fluoride (MgF_2_)-layers have been previously investigated for a broader range of applications, varying from orthopedic to cardiovascular and oropharyngeal applications [47,48,49]. Thereby, a generally higher corrosion resistance could be detected [37,47,48,49,50,51,52,53,54,55]. Therefore, the combination of a magnesium mesh embedded in a native collagen membrane allows volume-stable interventions and complete biodegradation.

Thus, the purpose of this study was the in vitro and in vivo investigation of a newly developed volume-stable and weight-bearing GBR/GTR-membrane. The combination of a native collagen membrane with an HF-treated magnesium mesh to control hydrogen evolution and the overall degradation process represents a novel approach of this research objective. To our best knowledge, this was the first in vitro/in vivo assessment of a HF-treated magnesium mesh integrated in a native collagen membrane for GBR/GTR therapy. These may preferably be used in trauma surgery as well as oral surgery.

In vitro, both HF-treated and untreated magnesium variants were tested for their basic cytocompatibility to ensure a safe in vivo application. Thereby, the HF-treated membrane exhibited promising cytocompatibility in the extract and direct tests, while the untreated membrane was within the toxic range in all extract assay. The HF membrane was also toxic in the LDH assay. However, the difference with the negative control was not significant, which in turn indicates only a slight toxicity. After 72 h extraction, the untreated magnesium showed clear signs of corrosion, while the HF-treated magnesium showed no macroscopic changes of the surface. This result supports the observation that HF-layers protect magnesium from premature degradation and retard the precipitation of corrosion products. In the direct tests, both magnesium variants showed many living cells with better cell attachment for the HF-treated magnesium. Due to the good results in the direct test, the untreated membrane qualified for further in vivo tests. In concordance with other in vitro studies, the results also showed that HF-treatment of magnesium leads to improved cytocompatibility, cell attachment and corrosion resistance [37,48]. Overall, the fabricated prototype was elastic and adaptable to the defect, so that volume stability can be insured. Thus, the new membrane exhibits similar properties as the already available non-absorbable PTFE membranes with integrated titanium meshes. However, further studies should investigate the applicability under real treatment conditions.

In vivo, 18 New Zealand White Rabbits were randomly divided into four study groups and three time points (6, 12, 18 weeks) as follows: (a) HF-treated magnesium mesh embedded in a native collagen membrane, (b) untreated magnesium mesh embedded in a native collagen membrane, (c) the native collagen membrane alone and (d) an empty control. The results of the study showed that total bone regeneration was similar for all groups at all time points, which questions the general necessity for GBR/GTR membranes. In this context, other studies discussed and showed similar results for uncovered bony defects compared to collagen membrane covered defects [56,57,58,59]. Moreover, the used animal model might be a reason for the comparable values as it does not reflect the clinical situation that is associated with the application of such a biomaterial. In this context, Byun et al. evaluated a hydroxyapatite (HA)-coated magnesium mesh in a rat calvarial model in a comparable experimental setup [60]. In terms of new bone formation, also no differences between all groups were found in the work of Byun et al. [60]. Regarding the microscopic appearance of the newly formed tissue, HA-coated meshes resulted in well-organized defect areas, while the empty controls seemed to be curved, distorted and filled with malformed collagenous tissue. However, such qualitative differences were not visible in our study, which may be due to the use of another animal model or the use of collagen for embedding [60]. Altogether, these qualitative results of the present study are subjected to limitations, as the application of the new membrane and the control groups should be analyzed in a load-bearing defect model in combination with and without bone substitute materials. Thus, an animal model should be applied, which resembles human conditions.

Moreover, the values for bone regeneration were different for the applied measurement methods according to the order of DVT > contact radiography > histomorphometry. These results may be explained by the different resolution capacities of the applied imaging procedures. Additionally, it has to be mentioned that both DVT and contact radiography allow for analysis of the complete defect sides, while the histomorphometrical method only enables for analysis of a representative 2D slide from the defect area. Since magnesium and bone have similar densities, different distinction properties of the applied measurements methods between bone tissue and the magnesium implants might be a further reason for the different results [61].

The histopathological observations in combination with the histomorphometrical analysis of the magnesium-associated gas cavity evolution are the most interesting results of the present study, even in view of the applicability of HF-treatment for the adaption of the degradation behavior of magnesium implants. The histopathological analysis showed that the HF-induced surface coating prevented gas cavity development up to 12 weeks compared to the values found in the group of the untreated magnesium implants, while these effects were no longer detectable after 18 weeks. Similar observations on delayed gas release of MgF_2_-treated materials could be shown in other studies as well [53,54,55]. Thereby, gas bubble release delayed between 90 and 180 days [53,54,55]. Compared with other coating technologies, HF-treatment can partially prevent a premature burst of major gas bubble formation [30,62]. However, the applicability for larger bone defects or fracture plates that require mechanical strength for about 3 months must be critically discussed and tested [63].

Moreover, the histopathological analysis revealed that the HF-treated magnesium meshes induced a layer of mononuclear phagocytes up to 12 weeks, in combination with a slight fibrosis involving mostly fibroblasts, while the untreated magnesium meshes only induced a fibrosis-like tissue reaction. After 18 weeks, the tissue reaction in both groups has shown to be similar. These result initially lead to the conclusion that the surface treatment that has already been shown to lead to the formation of a magnesium fluoride MgF_2_-layer onto the surface of such kind of implants has been phagocytosed within 12 weeks after implantation but enables to delay the hydrogen release and thus the gas cavity development.

In this context, in the work of Byun et al., no hydrogen gas release could be detected, most likely explained by the different coating or by rapid encapsulation and integration of the HA-mesh into the surrounding tissue [60]. In both studies, the implantation of the magnesium membranes triggered an inflammatory tissue reaction, being described as exaggerated by Byun et al. after 18 weeks [60]. Other studies described mild or unspecific inflammatory responses to HF-treated magnesium [53,54,55,64]. In the present study, both the HF-treated and the untreated magnesium meshes only showed minor inflammation, which decreased over time and seems to be driven by mononuclear phagocytosis for the MgF_2_ layer. Unfortunately, the cell composition involved in decomposition of the mesh was not further described by Byun et al., or other studies, and no comparable studies have been published until now dealing with this scientific topic [53,54,55,60,64].

Furthermore, the present observations lead to the conclusion that the hydrogen gas cavities are covered with fibrotic tissue. In this context, it has already been discussed that a biomaterial-related fibrosis might be a restricting factor for its regenerative capacities due to its isolation from the surrounding tissue [65,66,67,68,69]. However, other study results refute this assumption as it has been shown in the case of bone substitute materials, which partially also induce slight fibrotic encapsulation, that this tissue reaction does not interfere with the biomaterial-associated healing processes [70,71,72]. The latter is evidenced by the observation in the present study that—despite the observed different tissue responses—the magnesium meshes in both study groups were in close contact to newly formed bone tissue and, thus, did not interfere with the formation of new bone tissue.

This is the first study that provides results of magnesium-based meshes embedded in a collagen membrane for volume stable GBR/GTR therapy. Both in vitro and in vivo results are congruent with respect to improved cyto- and biocompatibility of HF-treated membranes. Nevertheless, this study and its results are subjected to limitations and further questions appeared. In vitro, corrosion measurements by potentiodynamic polarization or in a corrosion testing device can improve the results and the further applicability of the tested specimens. Further methods, such as hemocompatibility or tests for osteogenic differentiation, can be applied, as displayed in other studies of our group [29]. Furthermore, the mechanical characteristics of the MgF_2_ layer should additionally be tested before and during its application. In vivo, magnesium showed not to be fully degraded, which enquires the extension of the study period. A more detailed comparison of radiographic methods for measuring bone regeneration could provide further approaches to reduce in vivo studies. Closer examination of inflammatory relationships (e.g., the involvement of different cell types and the alignment of macrophages) could also be of interest for further development of this material class [19]. Finally, the direct comparison or amplification of other surface modifications such as duplex- (MgF_2_/PCL) [50], nanocomposite- [73] and composite-coatings [51,74] as well as the extension of other magnesium alloys [47,48,49,50,51,75,76,77,78,79,80,81] should be investigated and might promise new results in the future. Overall, especially in comparison with other studies, there is further need for simplification and standardization of in vitro and in vivo studies of magnesium-based materials.

## 4. Materials and Methods

### 4.1. Biomaterial Preparation

Magnesium sheets made of magnesium alloy AZ31 (biotrics bioimplants GmbH, Berlin, Germany) with dimensions of 30 × 40 mm were subjected for further processing. The samples were ultrasonically cleaned in 100% ethanol and distilled water. MgF_2_ coating was achieved by immersion of the test samples in 20% HF solution at 37 °C for 6 h and subsequently cleaned in 100% ethanol and distilled water. For the in vitro assessment, untreated and HF-treated sheets were used in dimensions of 0.5 × 0.5 cm^2^. For the in vivo assessment, untreated magnesium sheets were further processed using femtosecond laser cut to achieve a diamond pattern before treatment of the meshes by means of HF. Both treated and untreated magnesium meshes were embedded in a native porcine collagen membrane (Jason^®^ Membran, botiss biomaterials GmbH Berlin, Germany) that was further used as the negative control in vivo. Sterilization was achieved by gamma irradiation.

### 4.2. Cytocompatibility Analysis

Cytocompatibility analyses were executed in accordance with the ISO 10993-5/-12 and were already described in detail previously [29,40,63]. The experimental setup is described briefly in the following paragraphs.

#### 4.2.1. Reference Material

All reference materials were sterilized by immersion in isopropanol for 5 min with subsequent drying in a laminar flow hood. RM-A (Hatano Research Institute, Food and Drug Safety Center, Ochiai, Japan) was used as a positive control reference. Wako plastic sheets (Wako Pure Chemical Industries, Ltd., Osaka, Japan, Cat. No.160-08893) were used as a nontoxic control material. Samples of RM-A, Wako plastic sheets and titanium were prepared with the same surface areas as the material specimens and sterilized likewise.

#### 4.2.2. Cells and Cell Culture

Mouse fibroblasts of the L-929 cell line and mouse osteoblast precursor cells of the MC3T3 cell line were obtained from the European Collection of Cell Cultures, ECACC (Salisbury, UK). Cells were cultured in cell culture medium (MEM (Minimum Essential Medium)) supplemented with 10% fetal bovine serum, penicillin/streptomycin (100 U/mL each) (all from Life Technologies, Carlsbad, CA, USA) and L-glutamine (Sigma-Aldrich, St. Louis, MO, USA) to a final concentration of 4 mM under cell culture conditions (37 °C, 5% CO_2_, and 95% humidity). At about 80% confluency, cells were passaged.

#### 4.2.3. Extract Analysis

##### Extraction

Test and control samples were extracted for 72 h at a surface to volume ratio of 3 cm^2^/mL in cell culture medium under cell culture conditions. Cell culture medium alone was incubated under identical conditions to serve as a negative control extract. After removal of the specimens, the remaining extracts were centrifuged at 14,000 rpm for 10 min. The supernatants were used for the different assays that are described below.

##### Assay Procedure

In total, 96 well plates were seeded with 1 × 10^4^ L-929 cells/well in 100 μL cell culture medium and incubated under cell culture conditions for 24 h. Thereafter, cell culture medium was discarded and 100 μL of extract were added to each well. Cells were further incubated for 24 h and then subjected to the BrdU- and XTT-assays while the supernatants were subjected to the Lactate Dehydrogenase (LDH)-assay. Identical assays but omitting cells were conducted for all extracts as a control for assay interference. Blank controls (medium alone without cells) were subtracted from the absorbance values in all assays.

##### Bromodeoxyuridine/5-Bromo-2′-Deoxyuridine (BrdU)-Assay

A BrdU (colorimetric) test kit (Roche Diagnostics, Mannheim, Germany) was used according to the manufacturer’s instructions. Briefly, cells were labeled with BrdU for 2 h under cell culture conditions and subsequently fixed for 30 min at room temperature with a FixDenat reagent. Then, the fixed cells were incubated for 1 h with an anti-BrdU-peroxidase (POD) antibody and washed 3 times for 5 min with washing buffer. The immune complexes were detected after a subsequent substrate reaction with tetramethyl-benzidine (TMB) (20 min at room temperature) followed by the addition of 25 μL 1 M H_2_SO_4_ to stop the reaction using a scanning multi-well spectrophotometer (ELISA reader) with filters for 450 and 690 nm (reference wavelength).

##### Sodium 3,3′-[1(Phenylamino)carbonyl]-3,4-tetrazolium]-3is(4-methoxy-6-nitro) Benzene Sulfonic Acid Hydrate (XTT)-Assay

The Cell Proliferation Kit II (Roche Diagnostics, Mannheim, Germany) was used according to the manufacturer’s instructions. Briefly, the electron-coupling reagent was mixed with the XTT labeling reagent (1:50 dilution) and 50 μL of the mixture was added to the cells. After 4 h of incubation under cell culture conditions, substrate conversion was quantified by measuring the absorbance of 100 μL aliquots in a new 96 well plate using a scanning multi-well spectrophotometer (ELISA reader) with filters for 450 and 650 nm (reference wavelength).

##### Lactate Dehydrogenase (LDH)-Assay

The LDH-Cytotoxicity Assay Kit II (BioVision, Milpitas, CA, USA) was used according to the manufacturer’s instructions. Briefly, 10 μL of the cell supernatants were incubated with 100 μL LDH reaction mix for 30 min at room temperature. After the addition of stopping solution, absorbances were measured using a scanning multi-well spectrophotometer (ELISA reader) with filters for 450 and 650 nm (reference wavelength).

#### 4.2.4. Live-Dead Staining

Mg-test samples and controls were seeded with 2.4 × 10^5^ L-929-cells and 2.0 × 10^5^ MC3T3-cells in 1 mL medium in each well of 12 well plates (the surface area/medium ratio was 5.65 cm^2^/mL). Assays were carried out after 24 h incubation under cell culture conditions. In order to perform live-dead cell staining on the surfaces of the specimens, 60 μL per mL medium propidium iodide (PI) stock solution (50 μg/mL in PBS) and 500 μL per mL medium fresh fluorescein diacetate (FDA) working solution (20 μg/mL in PBS from 5 mg/mL FDA in acetone stock solution) were added to each well (12 well plate). After a brief incubation for 3 min at room temperature, specimens were rinsed in prewarmed PBS and were immediately examined with an upright fluorescence microscope (Nikon ECLIPSE Ti-S/L100, Nikon GmbH, Düsseldorf, Germany) equipped with a filter for parallel detection of red and green fluorescence. Pictures were taken using a 4×, 10× and 20× objective. Cells visualized using the 10× objective were counted using the software ImageJ. For each material, cells were counted from three experiments to calculate mean and standard deviation.

### 4.3. Experimental Animals and Surgical Procedure

Eighteen female New Zealand White Rabbits (Charles River Laboratories, Wilmington, MA, USA), randomly divided into three groups (6, 12, 18 weeks) weighing between 2.2–2.9 kg at the age of approximately 12 weeks, were used for the experiments. The rabbits were acclimatized for four weeks upon delivery, had ad libitum access to water and dry food, and were kept under a constant day-night cycle. This study was conducted at the Animal Research Laboratory of the University Medical Center Hamburg-Eppendorf by the Department of Oral and Maxillofacial Surgery and was approved by the Veterinary Office of the Hamburg Authority for Health and Consumer Protection (No. 29/16, approval date: 19 May 2016).

Prior to implantation, the animals were anaesthetized with 5 mg/kg xylazine (Rompun^®^ 2%, Bayer Vital GmbH, Leverkusen, Germany) and 70 mg/kg ketamine (Ketamin 10%, bela-pharm GmbH & Co. KG, Vechta, Germany). 11 animals additionally received buprenorphine (Buprenovet^®^, 0.3 mg/mL, Bayer Vital GmbH, Leverkusen, Germany) and 7 animals received Fentanyl (5–20 µg/kg/h) as an opioid. The calvariae of the rabbits were shaved and the skin was disinfected with Cutasept^®^ F (BODE Chemie GmbH, Hamburg, Germany). A transverse skin incision was made along the midline of the skull, and the periosteum was elevated laterally. Four circular defects, of each 8 mm in diameter were created (Figure 10) (Meisinger, Neuss, Germany, Ref. No. 330205486001070). After ensuring adequate hemostasis using electrocoagulation, each defect was either covered by a porcine collagen membrane (Jason^®^ Membran, botiss biomaterials GmbH, Berlin, Germany), a magnesium reinforced collagen membrane or a HF-treated magnesium-reinforced collagen membrane, respectively, in accordance with a randomized trial protocol. One defect remained empty and served as control.

After this insertion procedure, the periosteum layer and the muscle were relocalized and stitched with individual button sutures (4-0 Vicryl^®^, Ethicon Inc., Somerville, NJ, USA). Finally, the skin was closed with surgical staples and the surgical site was treated with a wound spray (Desitin^®^ Salbenspray, Desitin Arzneimittel GmbH, Hamburg, Germany). Each animal received 10 mg/kg enrofloxacin (Baytril^®^ 10%, Bayer Vital GmbH, Leverkusen, Germany) as perioperative antibiotic prophylaxis up to 5 days post-surgery.

Six rabbits each were sacrificed at each study time point, i.e., after 6, 12 and 18 weeks, by intracardial injection of 3–5 mL T61 (T61, MSD Tiergesundheit, Luzern, Switzerland) after general anesthesia by the abovementioned anesthetics. Immediately, the calvaria, liver, kidneys, and spleens and the calvarium containing the implants were harvested and immerged in a solution of 4% formaldehyde for 48 h.

### 4.4. Digital Volume Tomography (DVT) Analysis

Digital volume tomography (DVT) was carried out by a ProMax unit (Planmeca, Finland) using a voltage of 90 kV and an X-ray tube current of 10 mA. A three-dimensional reconstruction of the calvarial defects was created by the open source program “Horos” (https://www.horosproject.org) using a slice thickness 6.45 mm. The bone surface measurement was then performed by the open source program “ImageJ” (https://imagej.nih.gov/ij/) and was limited to a circular area of 8 mm in diameter, thus, the size of the defect created. The color threshold of bone density was set for each defect individually within the range of 73 to 137 using a color depth of 8-bit.

### 4.5. Radiological Analysis

Contact radiography was performed by an LX-60 (Faxitron X-ray LLC, Tucson, AZ, USA) with exposure parameters of 1000 ms at 26 kV. The ImageJ software was then used to perform bone surface measurements on the resulting TIF-file analogous to the DVT analysis.

### 4.6. Histological Work Up

After completion of DVT and radiological imaging, each defect was separated sagittally, resulting in two halves. Subsequently, dehydration via a series of increasing alcohol concentrations and a final exposure to methyl methacrylate (MMA), benzoyl peroxid (BPO) and nonylphenyl-polyethyleneglycol acetate were performed before paraffin embedding. Cross sections with a thickness of 35 µm were prepared by means of a Leica SM 2500E (Leica Microsystems GmbH, Wetzlar, Germany) and stained with Masson’s trichrome stain, Kossa van Gieson’s stain and Toluidine blue stain. Undecalcified thin section preparation (30 µm) was performed on the remaining half [82,83].

### 4.7. Histopathological Analysis

The histopathological analysis focused on the outcome of the material-mediated bone defect healing and the comparison of the tissue reactions to the different membrane materials based on a previously published protocol [19,41,42,43]. A light microscope Axio Scope.A1 (Carl Zeiss Microscopy GmbH, Jena, Germany) was used for the analysis. These analyses included the evaluation of the following parameters: fibrosis, hemorrhage, necrosis, vascularization and the presence of neutrophils, lymphocytes, plasma cells, macrophages and multinucleated giant cells (BMGCs). Microphotographs were taken with the cellSens Entry (Olympus K.K., Tokyo, Japan) software using a DP72 (Olympus K.K., Tokyo, Japan) digital camera connected to the microscope.

### 4.8. Histomorphometrical Analysis

The histomorphometrical analyses included the comparative measurements of the amount of regenerated bone, which was measured under 20× magnification using a Zeiss 47 52 69 light microscope (Carl Zeiss Microscopy GmbH, Jena, Germany) and a calibrated scale. Additionally, the size of cavities related to the magnesium grids were histomorphometrically measured. Briefly, so-called “total scans” were generated with the aid of a specialized scanning microscope, which consists of an Axio Scope.A1 (Carl Zeiss Microscopy GmbH, Jena, Germany) combined with a digital camera and an automatic scanning table (Märzhäuser, Wetzlar, Germany) connected to an PC system running the Zen Core software (Zeiss, Tokyo, Japan). The resulting images were composed of 100 to 120 single images with a 100× magnification in a resolution of 2500 × 1200 pixels and contained the complete implant area as well as the peri-implant tissue. For the histomorphometrical measurement of the cavities, the slides stained by Toluidine blue were digitized. These images allowed the measurements of the cavity sizes by means of the Zen Core software.

### 4.9. Statistical Analysis

Quantitative data were statistically analyzed via an analysis of variance (ANOVA) and a following Bonferroni post-hoc test via the SPSS software (SPSS 24, IBM, Armonk, NY, USA). Statistical differences were designated as significant if *p*-values were less than 0.05 (* *p* ≤ 0.05) and highly significant if *p*-values were less than 0.01 (** *p* ≤ 0.01). Finally, the data were shown as mean ± standard deviation.

## 5. Conclusions

HF-treated magnesium meshes embedded in native collagen-based barrier membranes represent a promising volume stable alternative for GBR/GTR therapy in comparison to conventional non-resorbable membranes. The material fulfills the requirements for cyto- and biocompatibility, but should be more intensively be investigated in further studies.

## Figures and Tables

**Figure 1 ijms-21-03098-f001:**
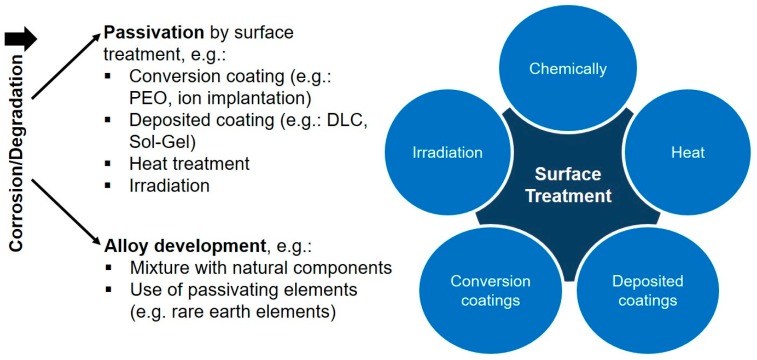
Common strategies to passivate magnesium for in vitro/in vivo application.

**Figure 2 ijms-21-03098-f002:**
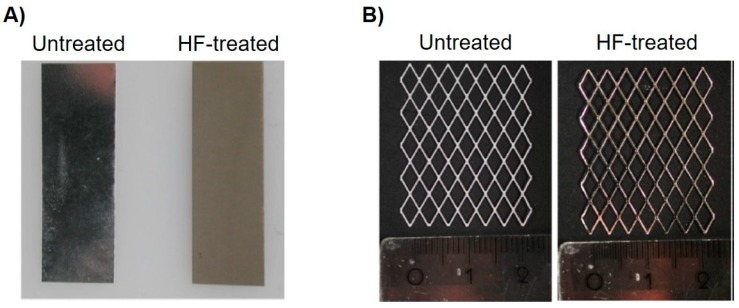
Comparison of untreated and processed samples. (**A**) Pictures of both an untreated and HF-treated magnesium sheet for the in vitro experiments. (**B**) Processed magnesium meshes before embedded into a collagen membrane.

**Figure 3 ijms-21-03098-f003:**
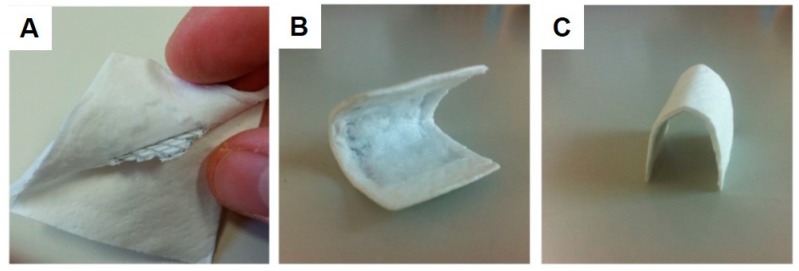
Different membrane modifications. (**A**) Integration of the magnesium mesh into the collagen fleece. (**B**,**C**) Moldability and stability of the collagen membrane with an integrated Mg mesh.

**Figure 4 ijms-21-03098-f004:**
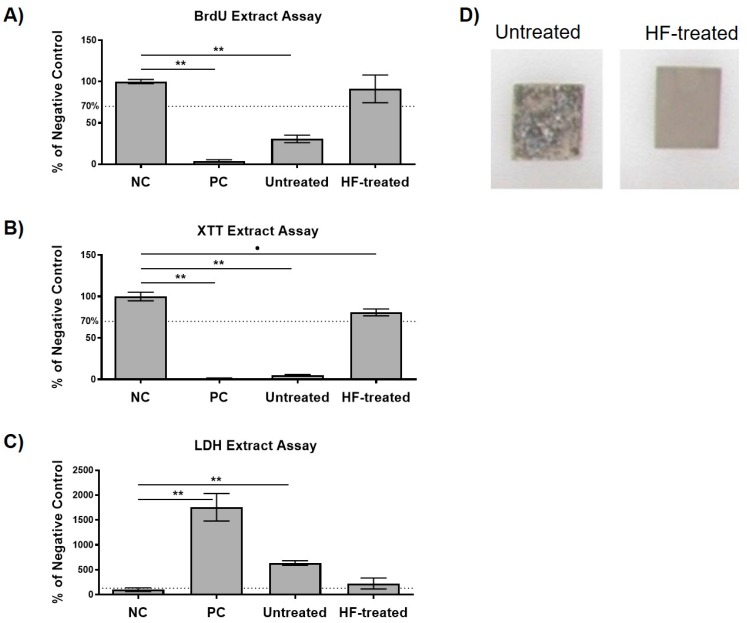
Cytocompatibility results using L9292 cells of the different variants. (**A**) proliferation measured by a BrdU assay; (**B**) viability measured by a Sodium 3,3′-[1(phenylamino)carbonyl]-3,4-tetrazolium]-3is(4-methoxy-6-nitro) Benzene Sulfonic acid Hydrate (XTT)-assay; (**C**) cytotoxicity measured by a Lactate Dehydrogenase (LDH) assay. Values are either normalized against positive controls (LDH) or negative control (XTT, BrdU). Means with error bars indicating standard deviations. The dotted line indicates thresholds which should not be exceeded (LDH) or fall below (XTT; BrdU). Significant differences are indicated (•: *p* < 0.05, **: *p* < 0.01). (**D**) Both untreated and HF-treated magnesium after 72 h extraction. The untreated magnesium shows enhanced corrosions, which is visible due to the high surface porosity and black textured corrosion products. The HF-treated magnesium was not different from the initial morphology.

**Figure 5 ijms-21-03098-f005:**
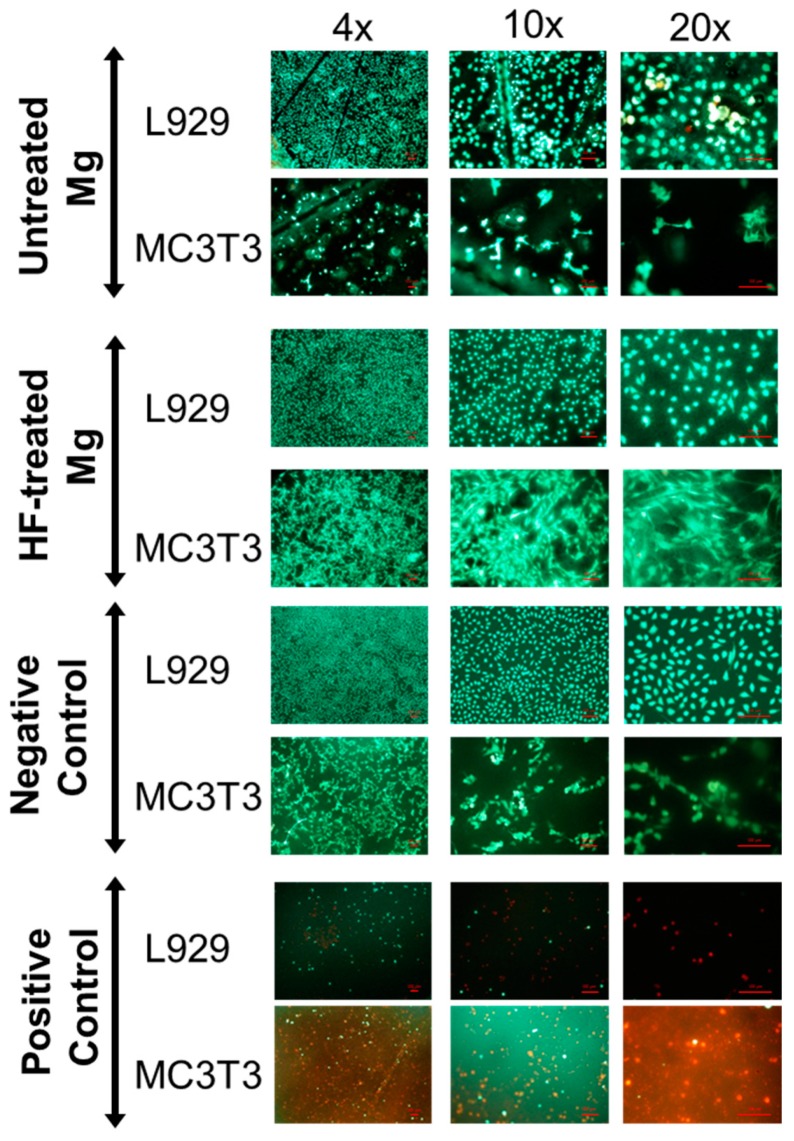
Attachment of cells on surfaces of the different variants and controls. The pictures show the attachment, vitality and morphology of the cells. Green: vital cells; red: dead cells. Spindle-shaped morphology indicates healthy cells with firm attachment. Round cells indicate poor attachment onto the surface.

**Figure 6 ijms-21-03098-f006:**
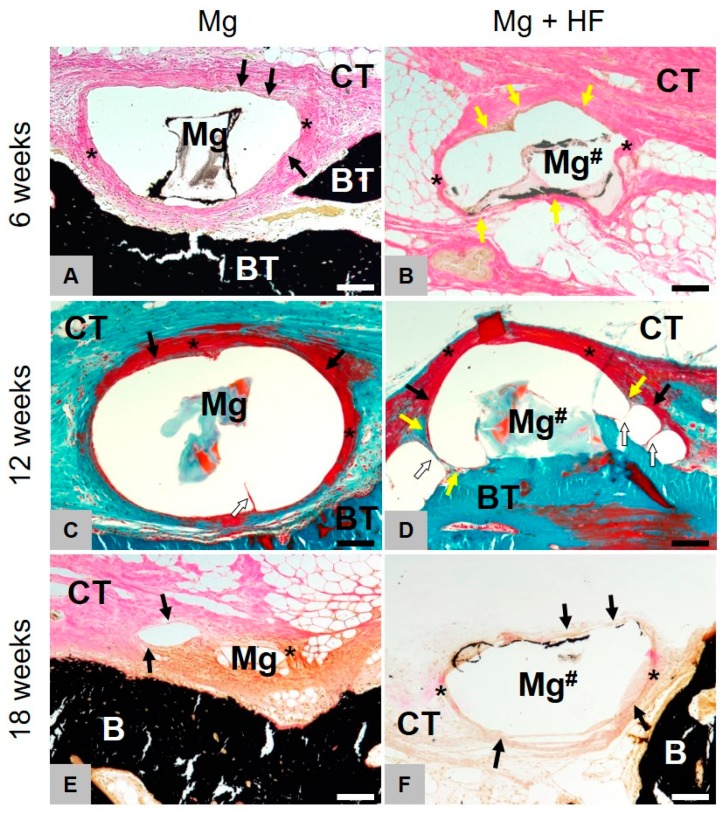
Histopathological comparison of both treated and HF-treated membranes. Images of Masson-Goldner (**C**,**D**) and Von Kossa (**A**,**B**,**E**,**F**) staining of the implantation site at 6, 12 and 18 weeks (100× magnifications, scalebars = 100 µm). Left for untreated (Mg) and right for HF-treated (Mg^#^) magnesium. Mg is mainly degraded via dissolution and scarcely through phagocytic processes. Mg-HF however, is primarily being resorbed via active phagocytosis and non-cellular dissolution only plays a minor role. After degradation of the HF-coating, decomposition, as with untreated Mg, principally occurs non-cellular-driven but through dissolution. Yellow arrows = phagocytic cells, black arrows = fibroblasts, asterisks = slight fibrosis, white arrows = septa between the gas cavities.

**Figure 7 ijms-21-03098-f007:**
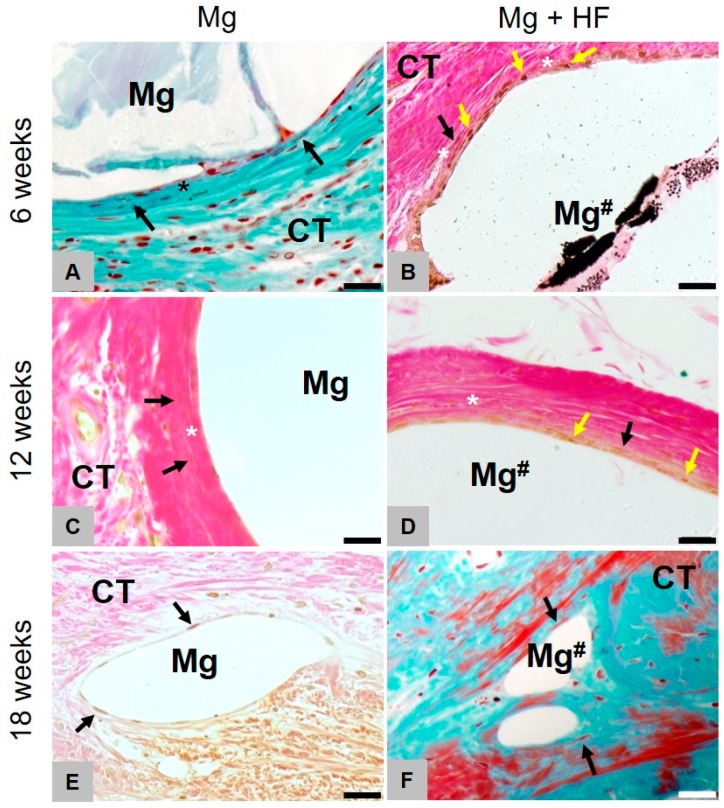
Gas cavity formation of HF- and untreated membranes.Representative images of Masson-Goldner (**A**,**F**) and Von Kossa (**B**–**E**) staining (40×) of the implantation site (scalebar = 20 µm) at 6, 12 and 18 weeks. Left for untreated (Mg) and right for HF-treated (Mg^#^) magnesium. Fibrotic capsule forming (*****) is visible at all times whereas gas cavity formation ceased to show after 12 weeks. The HF-coated mesh was mainly degraded by mononuclear cells (yellow arrows) up to 12 weeks, while the uncoated magnesium meshes elicited a fibrosis-like tissue reaction showing fibroblast accumulation (black arrows). Starting from 18 weeks after implantation, the tissue reaction in both groups was similar. Yellow arrows = phagocytic cells, black arrows = fibroblasts, asterisks = slight fibrosis.

**Figure 8 ijms-21-03098-f008:**
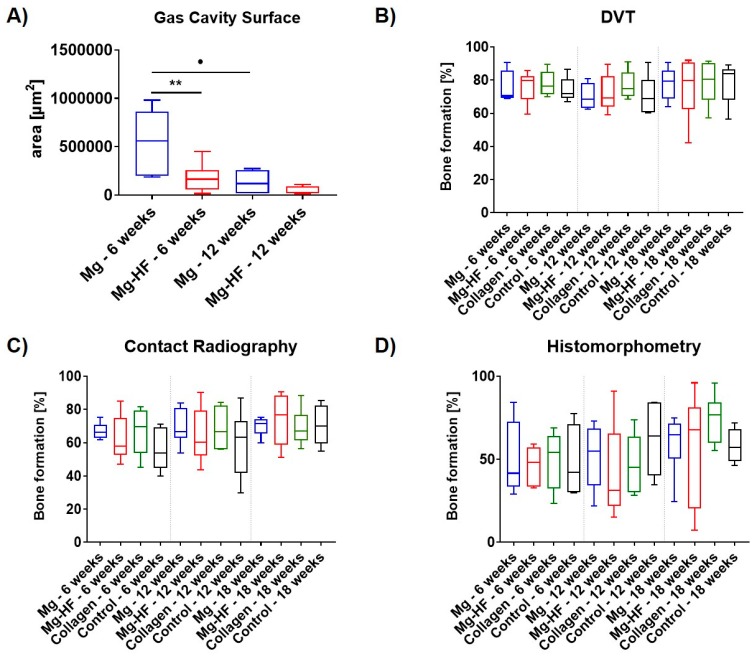
Overview of the various quantitative measurements. (**A**) Gas cavity surface in µm^2^ for uncoated (Mg; blue) and HF-treated (Mg-HF; red) magnesium after 6 and 12 weeks. HF-treated magnesium meshes show significantly lower gas cavity development compared to untreated magnesium meshes (**: *p* < 0.01) up to 6 weeks after implantation. Gas cavity dimension of untreated magnesium meshes shows to be reduced significantly after 12 weeks (•: *p* < 0.05). (**B**–**D**) Bone regeneration as measured with contact radiography, DVT and in histomorphometry after 6, 12 and 18 weeks for uncoated (Mg; blue) and HF-coated magnesium (Mg-HF; red), collagen (Collagen; green) and control (Control; black). No significant differences (*p* > 0.05) were detectable.

**Figure 9 ijms-21-03098-f009:**
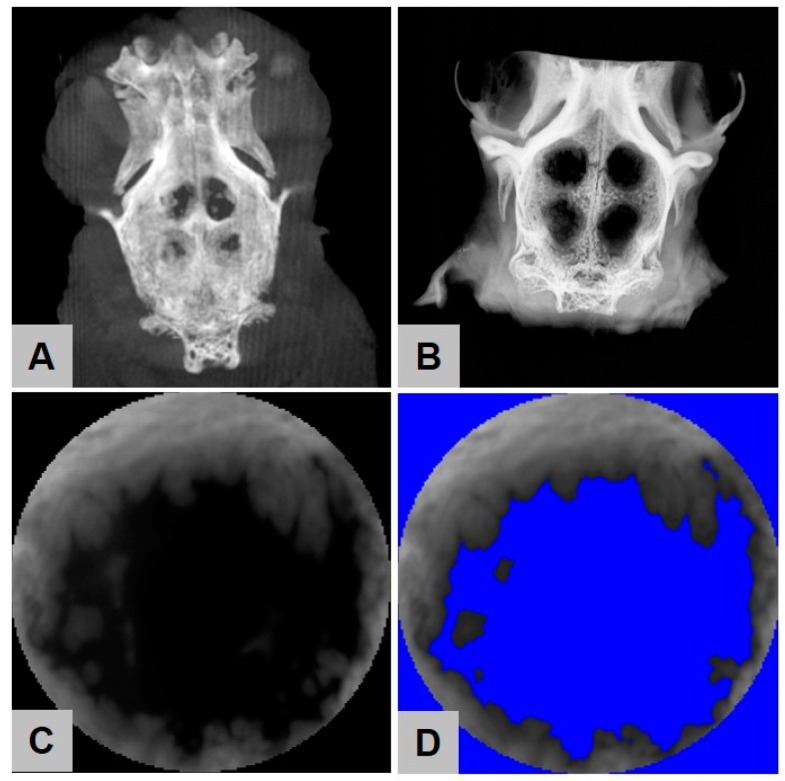
Imaging procedures to detect bone regeneration. (**A**) Three-dimensional DVT reconstruction and (**B**) radiological image as used for bone surface regeneration measurements. Selected circular defect area (**C**) before and (**D**) after application of color threshold.

**Figure 10 ijms-21-03098-f010:**
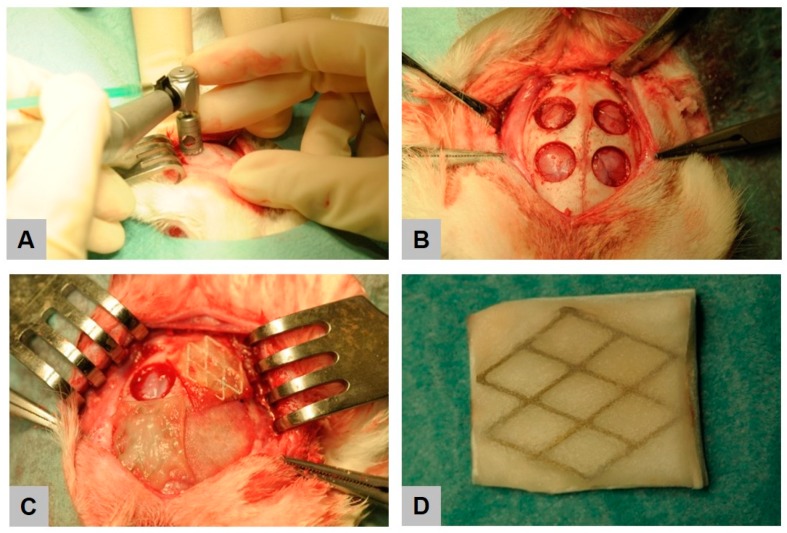
Intraoperative view on calvarial implantation model. Creation of four circular defects using a trephine burr (**A**,**B**), followed by placement of membranes (**C**). Image of the hydrofluoric acid (HF)-treated magnesium (Mg) mesh in a native collagen membrane **(D**).

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
