# Peer review of "Degradation, Bone Regeneration and Tissue Response of an Innovative Volume Stable Magnesium-Supported GBR/GTR Barrier Membrane"

_ijms, 2020, doi:10.3390/ijms21093098_

Round 1
Reviewer 1 Report
The manuscript describe in vitro and in vivo studies of a collagen/magnesium membrane fr bone regeneration, with an emphasis on the effect of HF treatment on the magnesium meshes.
Even though the study does not report significant improvement of outcomes in vivo, it provides important data that will allow future developments of these type of biomaterials.
Please see below for some minor corrections required:
The title should say GBR/GTR not GBT/GTR
There needs to be consistency in the way the samples are described. HF-treated would be better than HF-coated, MgF2-coated can be used as alternative. the term treated and untreated for the membrane should not be used as the meshes were treated not the membranes.
Section 2.1 please specify supplier and thickness of AZ31 sheets and also provide dimensions for the meshes.
Is the description of figure 2D correct?
In figures 7 and 8 make sure to specify what the arrows indicate and what the different symbols and abbreviations mean.
Careful proofreading is encouraged.
Author Response
Reviewer 1
The manuscript describe in vitro and in vivo studies of a collagen/magnesium membrane fr bone regeneration, with an emphasis on the effect of HF treatment on the magnesium meshes.
Even though the study does not report significant improvement of outcomes in vivo, it provides important data that will allow future developments of these type of biomaterials.
Please see below for some minor corrections required:
The title should say GBR/GTR not GBT/GTR
Responses: Corrected. Many thanks for the helpful comment.
There needs to be consistency in the way the samples are described. HF-treated would be better than HF-coated, MgF2-coated can be used as alternative. the term treated and untreated for the membrane should not be used as the meshes were treated not the membranes.
Responses: Corrected. Many thanks for the helpful comment.
Section 2.1 please specify supplier and thickness of AZ31 sheets and also provide dimensions for the meshes.
Responses: Corrected. Many thanks for the helpful comment.
Is the description of figure 2D correct?
Responses: Corrected. Many thanks for the helpful comment.
In figures 7 and 8 make sure to specify what the arrows indicate and what the different symbols and abbreviations mean.
Responses: Corrected. Many thanks for the helpful comment.
Careful proofreading is encouraged.
Responses: The manuscript was again proofreaded by a native speaker.
Reviewer 2 Report
In the materials and methods specify what the cells L929 and MC3T3 are.
Figure 3: please show a higher magnification to see the difference between treated and untreated scaffolds.
Figure 5: what type of cells are used ?
Figure 6: put a bar on each picture rather than magnifications 4X, 10X, 20X. After how long the live dead is realized?
Figure 7: what do the black and yellow arrows show. Remove 10X magnification. Give the meaning of the abbreviations : CT, BT, B. Describe (A, B, E, F) before (C, D). Why did you use two different stainings for different times. It would be better to show the 2 stainings for each time.
Figure 8: Remove 40X magnification, the bars are enough.
Figure 10: please show a higher magnification of the 4 lesions. After how long have you taken this photo? indicate which scaffold in each lesion. There seem to be differences between the 4 lesions???
Page 6 lanes 202-204: specify that there are also subcutaneous implantations.
Why did you choose the rabbit as a model?
Why did you implant scaffolds larger than the lesion?
Author Response
Reviewer 2
In the materials and methods specify what the cells L929 and MC3T3 are.
Responses: Corrected. Many thanks for the helpful comment.
Figure 3: please show a higher magnification to see the difference between treated and untreated scaffolds.
Responses: Many thanks for the useful comment. Unfortunately, no other images are available. We hope that the manuscript is nevertheless suitable for publication.
Figure 5: what type of cells are used ?
Responses: Corrected. Many thanks for the helpful comment.
Figure 6: put a bar on each picture rather than magnifications 4X, 10X, 20X. After how long the live dead is realized?
Responses: Many thanks for the useful comment. Unfortunately, we cannot change the images to include a scalebar any longer. We hope that the manuscript is nevertheless suitable for publication.
As also stated in section 2.2.4 the LIVE/DEAD-staining was conducted after 24h.
Figure 7: what do the black and yellow arrows show. Remove 10X magnification. Give the meaning of the abbreviations: CT, BT, B. Describe (A, B, E, F) before (C, D). Why did you use two different stainings for different times. It would be better to show the 2 stainings for each time.
Responses: Corrected. Many thanks for the helpful comment.
We used the different stainings for better illustration of the tissue reactions. We hope that the manuscript is nevertheless suitable for publication.
Figure 8: Remove 40X magnification, the bars are enough.
Responses: Corrected. Many thanks for the helpful comment.
Figure 10: please show a higher magnification of the 4 lesions. After how long have you taken this photo? indicate which scaffold in each lesion. There seem to be differences between the 4 lesions???
Responses: Many thanks for the useful comment. Unfortunately, no other images are available. We hope that the manuscript is nevertheless suitable for publication.
Page 6 lanes 202-204: specify that there are also subcutaneous implantations.
Responses: Corrected. Many thanks for the helpful comment.
Why did you choose the rabbit as a model?
Responses: The rabbit is a medium-sized experimental animal that is manifoldly used for analysis of the bony integration of such medical devices. We have very good experiences with rabbits as experimental animals for this model and analysis.
Why did you implant scaffolds larger than the lesion?
Responses: This was used to fixate the implants within their respective positions over the bony defects to prevent any disturbances.